

**Bipolar long-term high temporal resolution broadband**
**measurement system for incoming and outgoing solar UV**
**radiation, and snow UV albedo, at Sodankylä (67°N) and**
**Marambio (64°S)**
Meinander O[1], Aarva A[1], Poikonen A[1], Kontu A[2], Suokanerva H[2], Asmi E[1], Neitola K[1], Rodriguez
E[1], Sanchez R[3], Mei M[3], de Leeuw G[1,4] and Kyrö E[2]
[1] {Finnish Meteorological Institute, Helsinki, Finland}
[2] {Arctic Research, Finnish Meteorological Institute, Sodankylä, Finland}
[3] {Servicio Meteorologico National, Marambio, Argentina}
[4] {Department of Physics, University of Helsinki, Helsinki, Finland}
Correspondence to: Outi.Meinander@fmi.fi
**Abstract**
The polar regions of the Earth are characterized with low solar elevation angles, cold temperatures
and large amount of snow and ice. Under the harsh polar conditions, good quality *in situ*
measurements of incoming and outgoing solar radiation, and surface albedo, is a major challenge.
Both in the Arctic and Antarctic, the solar radiation at the wavelengths of ultraviolet (UV) has been
of particular interest due to the polar stratospheric ozone depletion, which increases UV radiation
on the ground (with multiple effects, such as UV-induced DNA-damage). In the presence of light
absorbing impurities (OC/BC/dust) in snow, albedo can be most affected at the wavelengths of UV.
Here we provide, for the first time, the description, comparison and evaluation of our bipolar
measurement design, instrumentation and data system, calibration assessment, as well as
challenges, for measuring incoming and outgoing UV radiation (from which snow albedo is
calculated) at the Sodankylä Arctic Research Centre (67°N) and at the Marambio Antarctic
Research Station (64°S). Both stations are members of the Global Atmosphere Watch GAW



programme of the World Meteorological Organization WMO and have personnel available year-
round. The Sodankylä snow UV albedo measurements were started in 2007, as part of the
International Polar Year IPY (2007–2008). The Marambio surface UV albedo measurements, since
February 2013, are part of the Argentinian-Finnish scientific co-operation on meteorological and
atmospheric observations. The paper aims to give a comprehensive insight into our bipolar
measurement system, and to create an understanding needed for a successful scientific utilization of
these data, including satellite and modeling approaches. We also present a literature review of our
previous publications on Sodankylä snow UV albedo data.

## 1 Introduction

The polar regions of the Earth are characterized with low solar elevation angles, cold temperatures
and large amount of snow and ice. Detecting changes in polar snow and ice albedo is of critical
importance for monitoring and modeling climate change, atmospheric chemistry, and satellite
applications. Under the harsh polar conditions, continuous good quality *in situ* measurements of
incoming and outgoing solar radiation present a major challenge.
The solar radiation reaching the ground surface at wavelengths of ultraviolet (UV) has been of
particular interest in both polar regions ever since the first observations of springtime losses of
ozone over Antarctica (Farman et al. 1985). The stratospheric ozone layer protects the life on Earth
from the Sun's harmful UV radiation by blocking all of the UV-C radiation, and most of the UV-B,
and about half of the UV-A. Also the solar elevation and clouds, air pollution, haze, and the
elevation above sea level affect the amount of UV radiation reaching the ground. The multiple
effects of UV irradiance include, for example, the harmful DNA damage (Sinha and Häder 2002)
and skin cancer, and the positive effects the production of vitamin D in the human skin. When UV
radiation is reflected from snow and ice, it may cause unprotected eyes the painful condition of
snow blindness (UNEP 2002). UV surface albedo is also an essential parameter for various RT
modeling applications, including satellite retrieval algorithms (Arola et al. 2003, Tanskanen and
Manninen 2007). Furthermore, snow albedo can be most affected at the wavelengths of UV in the
presence of light absorbing impurities (Meinander et al. 2013).
Here we for the first time describe, compare and evaluate our successful long-term high-temporal
resolution bipolar measurement design, instrumentation and data systems, and calibration
assessment, on incoming and outgoing solar radiation, and snow albedo, at ultraviolet (UV)
wavelengths at the Sodankylä Arctic Station (67°N), and at the Marambio Antarctic Station (64°S).


The term "Arctic" is used to refer to north of 60° northern latitude, and "Antarctic" to south of 60°
southern latitude. The purpose of this work is to provide a scientific reference for the Sodankylä and
Marambio measurement systems. The data are planned to be provided in open access data bases, to
allow a wider utilization of the data, e.g., in the WMO Antarctic Ozone Bulletins. Hence, this paper
aims to provide a reference needed by the data users. Although the focus of this paper is not to
publish the existing bipolar data or their scientific analysis (such a paper is at planning stage), we
will give a short review of the existing publications on Sodankylä snow UV albedo data (Meinander
et al. 2008, Meinander et al. 2009, and Meinander et al. 2013). The authors are not aware of any
previous literature review on snow UV albedo, and this is the first time we present a review of our
Sodankylä snow UV albedo results. Also, this is our first paper to consider the Sodankylä incoming
irradiance as an independent data set (instead of albedo). We will also discuss the potential of these
bipolar data for modeling and remote sensing applications.

## 2 Materials and methods

### 2.1 The polar WMO GAW stations of Sodankylä and Marambio

Both the Sodankylä and Marambio stations belong to the Global Atmosphere Watch (GAW)
programme of the World Meteorological Organization (WMO) (www.wmo.int/gaw). The focal
areas of GAW are aerosols, greenhouse gases, selected reactive gases, ozone, UV radiation, and
precipitation chemistry or atmospheric deposition. Sodankylä is situated in the Finnish Lapland, and
Marambio is an Argentinian Antarctic Base (Fig. 1). We use the same type of broadband UV
instrumentation in Marambio and Sodankylä to allow scientific bipolar comparisons.

### 2.1.1 Sodankylä Arctic Research Centre

The Arctic snow UV albedo is measured at the Sodankylä Arctic Research Centre of the Finnish
Meteorological Institute (FMI), Finland (67◦22'N, 26◦39'E, 179 m a.s.l.). The area is north of the
Arctic Circle and belongs to the northern boreal forest zone, with the snow type of taiga. The
seasonal snow cover in Sodankylä lasts in average from 15 October to 14 May (FMI statistics for
1981-2010, www.fmi.fi). For solar irradiance, Sodankylä is characterized with solar extremes:
nights of the Midnight Sun (no Sun set) from 30 May till 15 July, and Polar Nights (no Sun rise)
from 19 to 25 December (www.sodankyla.fi). Sodankylä is often beneath the middle or the edge of
the stratospheric polar vortex, and in the zone of polar stratospheric ozone depletion.


We measure the UV albedo at the FMI Sodankylä Arctic Centre operational albedo field (Fig 2).
The size of the field is 16 m×16 m, and low metal fences protect the operational field to have as
untouched snow surface during snow-time as possible. The snow UV albedo measurements were
started in 2007, as part of the International Polar Year IPY (2007–2008) ORACLE-O3 cluster
project (Ozone layer and UV radiation in a changing climate evaluated during IPY, http://www.awi-
potsdam.de/atmo/ORACLE-O3, Meinander et al., 2008, 2009).

### 2.1.2 Antarctic Marambio Base

The Antarctic UV albedo is measured at the Argentinian Antarctic Marambio Base (64º14'S,
56º37'W, 198 m a.s.l.). The station is located in the ice-free Marambio Island, on the north-east side
of the Antarctic Peninsula. The ecosystem type is Permafrost. The Antarctic region has no
permanent human habitation, but the Marambio Base is manned all-year-round. Marambio is of
great significance to studies related to the Antarctic Ozone Hole because it is on the edge of the
polar vortex. The polar spring arrives in Marambio in September or October, with return of
sunlight. However, the frequency of cloud-free days is small throughout the year, on average only
0-1 from November to February, and the mean frequency of days with snow fall varies between the
maximum 16 days in January to 7 - 8 days in June-September; snowfall is most common from
October to March (SMN statistics for 1971-1990, www.smn.gov.ar). Antarctic Peninsula has
experienced warming at rates several times the global mean (Trenberth et al. 2007).
The continuous Antarctic snow UV albedo measurements were started in 2013, as an Argentinian-
Finnish scientific co-operation of FMI and SMN. The measurements were installed to a new
container (Figure 3). In addition to the continuous measurements of the incoming and outgoing
ultraviolet radiation, i.e. snow UV albedo, the container houses a new weather station, and
instruments to measure the optical properties, numbers and chemical composition of aerosol
particles, and the concentrations of carbon dioxide and methane. Due to the extremely windy
conditions in Marambio, the snow cover is sometimes blown away in one place revealing the bare
ground (also under the albedo sensors), while the wind may drift snow in other places. Therefore,
the surface photographs are essentially needed to evaluate the surface condition during the albedo
measurements (snow/no snow).  Also, the Antarctic Peninsula differs from most of Antarctica by
having a summer melting season. Summer melt can produce snow-free areas.

### 2.2 Bipolar broadband UV albedo





Snow surface broadband UV albedo in Sodankylä and Marambio is measured bi-hemispherically
using two SL501 (Solar Light Co.) radiometers in both places. This means a maintenance and
calibration need of four SL-501 sensors. For detecting albedo, one sensor is installed to face
upwards, and the other downwards. According to the WMO, the operational meteorological local
albedo is defined to be measured at a standard height of 1–2 m (WMO, 2008, I. 7). This defines the
measurement height of 2 m for Sodankylä and Marambio UV albedo measurement
In principle, such UV broadband radiometers have simple operational requirements. Yet, the
relationship between the raw signal and the UV radiation product requires characterization and
calibration procedure for each individual broadband radiometer (Hülsen and Gröbner 2007).
The well maintained and calibrated radiometer pairs are selected to represent as similar spectral and
cosine responses as possible, as demonstrated in Fig. 1 of Meinander et al. (2008). The radiometers
measure the incoming irradiance weighted with the action spectrum for ultraviolet induced
erythema (McKinlay and Diffey1987), which also has a contribution from the UVA. The
measurement data are recorded in 1 min intervals. Hence, the SL501 measured dose of the radiant
energy Qery is calculated as the temporal and spectral integral of the convolution of the global solar
spectral irradiance and the erythemal response, measured hemispherically at 2pi. For the
erythemally weighted broadband albedo Aery, the ratio of the hemispherically measured up-welling
(↑) to down-welling (↓) UV solar radiation is then calculated as:

$$Aery = \frac{\text{Qery}\uparrow}{Qery\downarrow}$$  (1)

where the measured downwelling incoming irradiance includes both the direct and diffuse
components, and the upwelling outgoing part consists of the hemispherically reflected global
spectral diffuse radiance.
**2.3 The data collection systems in Sodankylä and Marambio**
The data collection systems in Sodankylä and Marambio (Fig. 4) are similar to each other. In
Sodankylä, the UV-Albedo data collection system measures and saves data automatically in one
minute interval. The system consists of sensors (2 x SL501A), loggers (Vaisala QML201), serial
device servers (NPort 5150A) and an embedded linux computer (Moxa IA240). The logger
measures raw data every 5 second for each sensor. Then the logger calculates one minute average





values and sends them to the linux-computer via the serial device server. The linux-computer sends
then values to a database and then also saves them to a dayfile on an SD-card. All the data are
available in the FMI Climate data base.
Similarly to Sodankylä, the UV-Albedo data collection system in Marambio measures and saves
data automatically in one minute interval. The system consists of sensors (2 x SL501A and 1 x
CMP11), a logger (Vaisala QML201) and an embedded linux computer (Moxa IA240). The logger
measures raw data every 5 second for each sensor. Then the logger calculates one minute average
values and sends them to the linux-computer. The linux-computer saves values to a dayfile on an
SD-card. The SD-card is copied manually by the system user. The data are then sent to FMI by
email, and stored in a server.

### 2.4 Calibration of the radiometers

The four SL501 sensors needed for the purpose are maintained by FMI. The sensors are sent for
calibration and for cosine and spectral response characterization to the Finnish Radiation and
Nucleation Safety Authority (STUK). STUK determines the calibration factor (C) for each SL501
sensor. The measured 1-min values are corrected by the calibration factor to gain the final
measurement data. More details can be found in Meinander et al. (2008). The STUK procedure is
the same for all the sensors used here.
At Sodankylä, the pair of sensors is changed each year to another pair of calibrated and maintained
sensors. In Marambio, the sensors were changed to a new pair of calibrated and maintained sensors
after the first year of measurements (2013). The current pre-calibrated Marambio SL501 sensors
have been in use for 2014-2015. A post-calibration will be made when the Marambio sensors are
transported back to Finland.

### 2.5 The biggest challenges in operating the bipolar measurements

The major general challenges related to our bipolar measurement data, according to our experience,
are listed in Table 1. These include the general challenges of: i) *Temperature effect*: all the sensors
are temperature controlled; the temperature of the sensor needs to be controlled and monitored,
because SL501 measurement values can be temperature affected; in the data files one column
contains the sensor temperature recorded every minute, and these data are essentially needed in the
QA/QC of the data; and ii) *Cosine error*: the opening angle of the SL501A is slightly restricted due
to the sensor design, where the detector is situated lower than the filters (this is evident in the cosine
responses of the sensors, too). This can have effects especially in polar region, characterized with



low solar elevation angles. As a consequence of the cosine error effect, some data at low solar
elevation may be needed to be excluded from the scientific utilization.
In addition, at Sodankylä the biggest challenges in operating the measurements have been: i) *Wind:*
Due to the very mild wind, snow accumulates on the sensors; some frost can be formed; ii)
*Measurement horizon*: the operational albedo field is surrounded by trees. Especially due to the fact
that as Sun is really low during snow time, the tree shadows can be very long. The tallest
surrounding trees have been cut 19 October 2012 and that has been detected to cause a systematic
rise in the level of the reflected radiation.; iii) *Calibration and maintenance of the sensors and*
*operator access to the sensors*: The sensors are calibrated, maintained and placed in the albedo field
every spring when the Sun appears. To prolong the lifetime of the sensors, they are not kept outside
when the Sun is at lowest. Yet, there is already snow on the ground at the time of the installation,
and although as little disturbance to the snow surface is caused as possible, it is impossible to keep
the snow surface totally untouched during the installation. This may cause some error in the albedo
measurements until the snow surface is fixed with new snow.
At Marambio, there are no obstacles surrounding the sensor. In turn, the biggest measurement
challenges we have faced especially there have been: i) *Cleaness of the dome*: the surface of the
quartz domes are not easy to maintain cleanest and free of dust, as the shelter is visited once a week.
ii) *Wind and dust:* the dome may get scratches due to the ice, sand and dust drifting with the wind.
iii) *Access* to the stand of the both sensors is uncomfortable. iv) *Wind:* As result of the high winds
(50/70 Kt) can break the connector or produce radial shake, which may affect the levelling among
other things. v) *Wind*: snow is sometimes blown away in one place revealing the bare ground (also
under the albedo sensors).

**3 Scientific utilization of Sodankylä and Marambio data**
The purpose of the current paper is to document the materials and methods of the measurements to
ensure a successful scientific utilization of the data in the future, and provide a literature reference
for the measurements setup to be used, when the data are utilized. Hence, the focus of this paper is
not to publish the existing data nor their scientific analysis. Although we do not present
measurement data as such, we will give a short review of the scientific utilization of the snow UV
albedo data (Chapter 3, Table 2), and discuss some future possibilities (Chapter 4).
The Sodankylä and Marambio measurement data consist of independent data of incoming and
outgoing UV radiation, and snow UV albedo during snow time. The incoming solar irradiance



contains information on the state of the sky, and the outgoing reflected part about the surface
properties. Therefore, in addition to the measurement challenges (see Chapter 2.5), there are several
environmental factors that need to be considered in the scientific usage of the data. For example,
snow albedo increases as the solar elevation decreases, i.e., albedo is SZA dependent.
The snow/ice properties affecting the albedo of clean and dirty snow are discussed in detail in
Warren and Wiscombe (1980) and Wiscombe and Warren (1980). Albedo for clean snow at UV is
expected to be 0.97-0.99, but albedo for dirty, melting or optically opaque snow can be significantly
lower. Snow albedo varies with wavelength, and depends on a number of factors, such as the depth
and age of the snow cover, snow grain size, solar zenith angle, and cloud cover. The BC snow
albedo effect is the bigger the smaller the wavelength, i.e. many impurities absorb UV radiation.
The scientific utilization of Sodankylä UV albedo data has revealed lower albedo values (0.5-0.7)
than expected (~0.9) (Meinander et al. 2008 and 2013). Our preliminary analysis using Marambio
data, in turn, has revealed highly variable snow albedo values, due to temporally and spatially
changing snow conditions. Therefore, the scientific utilization of the Marambio data demands
surface photographs to be combined with the measured albedo data.
*Analysis of the first Sodankylä UV albedo data*
As soon as the Sodankylä UV albedo measurements were started in spring 2007, a comprehensive
analysis of Sodankylä data was made (Meinander et al. 2008). Ancillary meteorological automated
weather station (AWS) data, and other available ancillary data, were used to investigate the various
environmental factors related to the measured albedo signal. During the snow accumulation, the
erythemal UV albedo was found at midday to be A=0.6-0.8. During melt it was A=0.5-0.7,
respectively. The snow albedo showed an unexpected diurnal decrease of 0.05 after midday, during
the period of snow melt, when the sky was almost clear sky or with variable cloudiness. The decline
recovered later, and it was solar azimuth angle asymmetric. This might indicate a change in snow
properties. Two different independently measuring instruments confirmed this finding. In the
following mornings, snow surface temperature was < 0 ∘C. The diurnal change appeared for one to
two hours. Surface snow daily metamorphosis can explain this. If the surface temperature increases,
it can melt snow. Later, the surface would freeze.
*Empirical parameterizations*
The parameterizations obtained from the measurement data can be simple, yet useful. Using the
data, we have gained an empirical relationship for Sodankylä snow grain diameter (D) as a function





of day of year (t), and the daily maximum air temperature (Tmax). We have also the empirical
relationship of albedo as a function of the height of the snowpack (Meinander et al. 2008).
***Bipolar comparison of Sodankylä-Neumeyer data and physical reasons for the detected snow***
***albedo SZA asymmetry***
Soon after the Sodankylä snow UV albedo meaurements were started, Sodankylä data were
compared with the German Antarctic Neumayer Station (70°39'S, 8°15'W) UV albedo data
(Meinander et al. 2009). The same type of sensors, Biometer Model 501 from Solar Light Co.
(SL501), were used.
In the Arctic, a 10 % decrease in albedo as a function of time within a day was found. The albedo
changed from 0.77 to 0.67. In the Antarctic, the change was from 0.96 to 0.86. This is named as
"snow albedo SZA asymmetry". It means that snow albedo is different for the same SZA,
depending of the time, and that is not due to changes in the irradiance (due to, e.g., cloudiness
conditions). The asymmetry cases were analyzed according to meteorological data (air/surface/dew
point temperature, and relative humidity). We found physical explanations for our observations.
These were: 1. previous night's low surface temperature combined with high relative humidity. This
is favorable to frost. As a result, a higher albedo follows the next morning. 2. Previous night's snow
fall with higher reflectance. 3. Day time snow melt followed by night time refreeze.
We also listed there other possible sources of uncertainty, in the determination of the SZA
asymmetry: i) the improper leveling of the sensor; ii) uneven snow surface (e.g., Antarctic sastrugi);
iii) shadowing due to objects (containers, houses, trees, slopes) in the vinicity of the measurement
site; iv) the different cosine behavior in different azimuthal planes of the diffusers. The detected UV
albedo decline in Neumeyer data was ($c$ = -0.024):

268        *$A = -0.0024*SZA$.*                                          *(2)*


Eariler, Pirazzini (2004) suggested for Antarctic snow (using another independent data set) a SZA
dependent snow albedo decline. The best afternoon fit for $c$ was -0.003. We can conclude the value
of $c$ to be surprisingly similar in our independent data set.
***Snow UV albedo at Sodankylä versus clean Arctic snow at 87°N***



In Sodankylä, snow UV albedo values have been lower than expected. In Meinander et al. (2013),
we found that in Sodankylä, albedo may be affected by high concentrations of BC, due to air masses
originating from the Kola Peninsula, Russia. Mining and refining industries are located there (Fig. 9 in
Meinander et al. 2013). As a comparison in clean Arctic environment, during the Arctic Ocean
ASCOS-expedition (Finnish contribution to the Arctic Summer Cloud Ocean Study), we measured,
using a NILU-UV radiometer, albedo values A = 0.91–0.92 for UV and PAR at 87°N (Paatero et al.

280  2009).

*Sodankylä snow UV albedo and organic and black carbon*
As part of the SNORTEX-2009 experiment in Sodankylä, snow albedo values were measured by
three different independent measurement set ups (Meinander et al. 2013). In addition to the SL-501
UV albedo, a double monochromator spectroradiometer Bentham for UV albedo, and pyranometer
setup for VIS albedo were used. UV albedo (at 330 nm) was detected to decrease from 0.65 to 0.45,
while most intensive snow melt took place. At visible wavelengths (450 nm), albedo changed from
0.72 to 0.53. Low albedo was confirmed by three independent simultaneous measurement devices.
We explained the low albedo values to be due to 1. large snow grain sizes (with diameter of 3 mm).
2. Meltwater increasing the effective grain size. 3. Light absorbing impurities (LAI) in the snow. At
the time of albedo measurements, snow contained 87 ppb of elemental carbon (black carbon, BC).
The organic carbon concentration was 2894 ppb. EC/OC analysis were performed using the
thermal–optical method. It was shown, that high EC in snow was due to air masses originating from
the Kola Peninsula, Russia. The place is known for mining and refining industries.

**4 Modeling and remote sensing applications**
**4.1 Snow UV albedo and BC: soot in snow absorbs the most at UV**
The black carbon (BC) has been estimated to be the second most important human emission after carbon
dioxide, in terms of its climate forcing in the present-day atmosphere (Bond et al. 2013). The effect on
reflectance of BC deposited on snow surface is the bigger the smaller the wavelength, i.e. the albedo effect of
BC is the biggest at UV (Fig 5, and Fig. 10 in Meinander et al. 2013). Based on the Soot on Snow
experiment (SoS-2013) in Sodankylä, we have also recently presented a new hypothesis (Meinander et al.
2014b) that soot may decrease the melt water retention capacity, and thus decrease the snow density of
melting snow.
**4.2 Satellite applications**



Satellite programmes produce measurements of atmospheric compounds and related parameters that
can be used together with the GAW network measurements. Putting together the highly accurate
local measurements from GAW ground-based stations and the statellite measurements with larger
coverage, a more complete picture can be achieved. The Committee on Earth Observation (CEOS)
has developed a strategy for such co-operation within an integrated system for monitoring of the
atmosphere (WMO 2001).
UV surface albedo is also a key parameter for applications using RT modeling, such as satellite
retrieval algorithms (Arola et al., 2003, Tanskanen and Manninen, 2007). The measurement set up
allows an independent use of the incoming and outgoing UV radiation. Therefore, the irradiance
data can help filling gaps in the incoming UV radiation measured within the Argentinian-Spanish-
Finnish co-operative NILU-UV Antarctic network (Redondas et al. 2008). Previously, those NILU-
UV data were compared with the overpass OMI UV satellite data (see Meinander et al. 2014a), and
the current data serves equally well for the same purpose.

**5 Conclusions**
We have successfully operated broadband radiometers for measuring incoming and outgoing
ultraviolet radiation in the harsh polar environmental conditions of Sodankylä, north of the Arctic
circle, and Marambio in the Antarctic Peninsula. Here we described the design and operation
demands from the bipolar point of view. The measurements  in Sodankylä were started as part of
the International Polar Year. The Marambio Antarctic measurements are applied for investigating
the incoming radiation, ground albedo during snow and no snow, and have a great potential to serve
satellite applications, too. Another important application may be expected from the fact that light
absorbing impurities deposited in snow can have bigger absorbance at the wavelengths of UV.
In the polar regions, the monitoring of incoming ultraviolet radiation is of special interest due to its
connection to stratospheric ozone depletion (often called the "ozone hole area", when total ozone is
less than 220 DU). The year 2015 has the third largest ozone hole on record according to this
criterion (WMO 2015a and 2015b), showing that the importance of ozone depletion is still actual.
Here we document, for the first time, the practical challenges in operating these measurement
systems in Sodankylä and Marambio. These challenges need to be carefully taken into account
when utilizing the data.





The data have been agreed to be submitted to the WMO GAW World Ozone and UV Radiation
Data Centre WOUDC (http://www.woudc.org/), and to the EUVDB database
(http://uv.fmi.fi/uvdb/), where they will be freely available for further scientific utilization.

**Acknowledgements**
We want to thank the personnel at Sodankylä FMI-ARC and Marambio base for their valuable
work. We gratefully acknowledge financial support from the Academy of Finland (projects
SAARA, A4, and ACPANT), and the Finnish Antarctic Research Program FINNARP.

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




Table 1. Summary of the challenges of the bipolar UV radiation measurements in Sodankylä
and Marambio.

| Challenge | Sodankylä | Marambio |
|---|---|---|
| Temperature | The temperature of the sensor needs to be controlled and the recorded sensor temperature to be used in the QA/QC of the data. | The temperature of the sensor needs to be controlled and the recorded sensor temperature to be used in the QA/QC of the data. |
| Low solar elevation | Cosine error of the sensor may cause a need to exclude low elevation data from the analysis | Cosine error of the sensor may cause a need to exclude low elevation data from the analysis |
| Wind | Due to the lack of wind, snow accumulates on the sensors; some frost can be formed. | As result of the high winds (50/70 Kt) can break the connector or produce radial shake, which may affect the levelling among other things. |
| Maintanence | Snow surface is not totally untouched during the installation. This may cause some error in the data until new snow. | The surface of the quartz domes are not easy to maintain cleanest and free of dust, as the shelter is visited once a week. |
| Measurement horizon | Trees surround the area. As Sun is low during snow time, the tree shadows can be long. Surrounding trees were cut  19 Oct 2012. | Not a problem, the horizon is open. |
| Dust | Not a problem. | The dome may get scratches due to the ice and sand and dust; surfaces of the quartz domes are not easy to maintain cleanest and free of dust. Snow can sometimes be dirty. |
| Snow surface | Seasonal snow cover, snow height varies. | Snow is sometimes blown away revealing the bare ground under the downlooking sensor |





Table 2. Summary of Arctic Sodankylä snow SL-501 UV albedo findings, and comparison results.

| Snow UV albedo findings | Reference |
|---|---|
| Sodankylä midday erythemally weighted SL-501 UV albedo was 0.6 - 0.8 in the accumulation period, and 0.5 - 0.7 during melt. | Meinander et al., (2008) |
| Sodankylä daily SL-501 UV snow albedo as a function of snow height ($h$) for 56<SZA<60 degrees during the melt $A=-6E-05h^2+0.0114h+0.1809$. | Meinander et al. (2008) |
| Diurnal decrease of 0.05 in Sodankylä snow SL-501 UV albedo soon after midday, and recovery thereafter, possibly due to snow metamorphism. | Meinander et al. (2008) |
| SZA asymmetry in snow SL-501 UV albedo: Up to 10 % decrease in albedo as a function of time within a day, ranging from 0.77 to 0.67 in the Arctic Sodankylä, and from 0.96 to 0.86 in the Antarctic Neumeyer station. Physical snow property related explanations given. | Meinander et al. (2009) |
| Arctic cleans snow, comparison NILU-UV snow albedo measurements at 87 deg N, A = 0.91–0.92 for UV and PAR. | Paatero et al. (2009) |
| Sodankylä melting snow low albedo values (Bentham~0.5–0.7, SL-501 ~0.4–0.5, and CM14 ~0.6–0.75) can be explained by 1. large snow grain sizes up to ~3 mm in diameter; 2. meltwater surrounding the grains and increasing the effective grain size; 3. absorption caused by impurities in the snow, with concentration of elemental carbon (black carbon) in snow of 87 ppb, and organic carbon 2894 ppb, at the time of albedo measurements. | Meinander et al. (2013) |
| Sodankylä snow albedo: Empirical albedo conversion between visible and UV albedo: A(VIS)=1.1602A(UV-B)−0.0213, (R2=0.6012). | Meinander et al. (2013) |
| 10 % daily melt time asymmetry in Sodankylä snow UV albedo. | Meinander et al. (2013) |




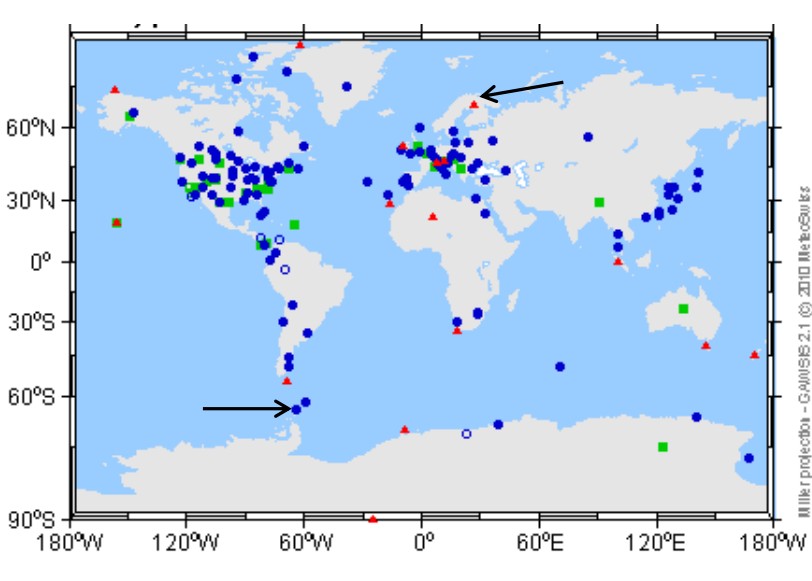



Figure 1. The UV radiation measurement stations of the WMO GAW network; Sodankylä and Marambio stations are indicated with black arrows (map downloaded using http://gaw.empa.ch/gawsis/).



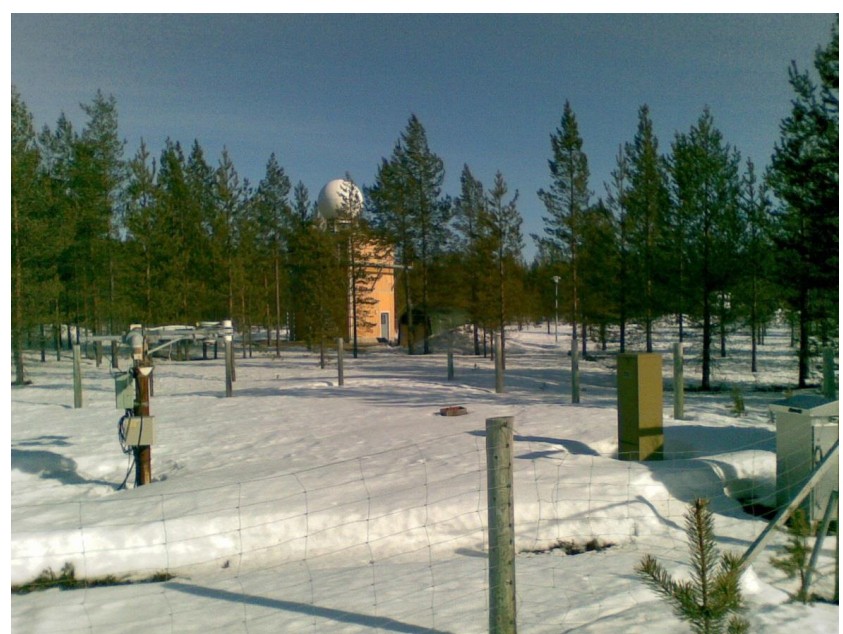



Figure 2. Sodankylä albedo field (16 m x 16 m) protected with fences and surrounded by pine trees.
The albedo sensors (on the left) are attached to a pole facing to the South.



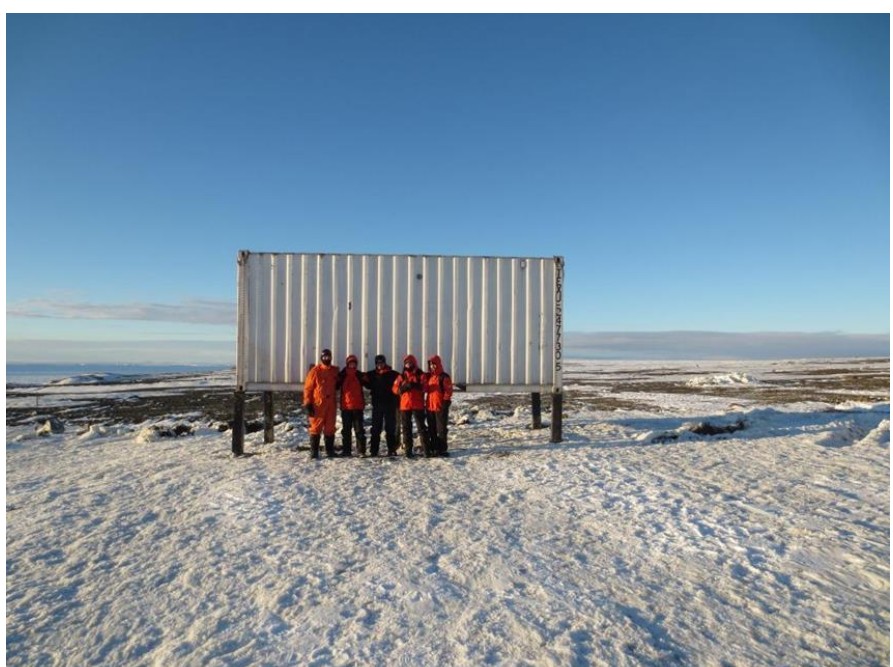

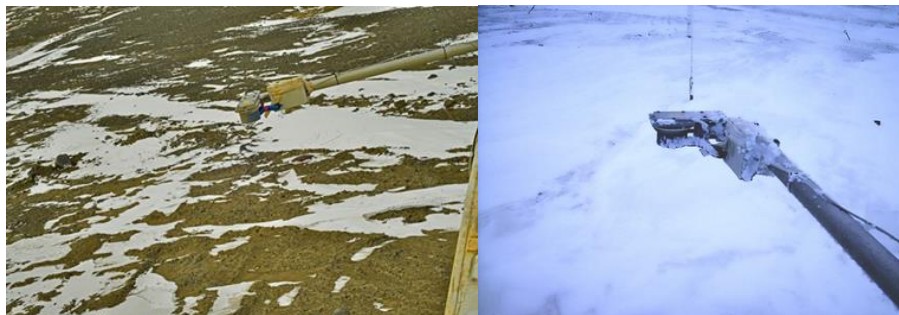


Figure 3. Marambio UV albedo measurement place and the environmental conditions for the
measurement of outgoing radiation.





## The block diagram of UV -Albedo measurements

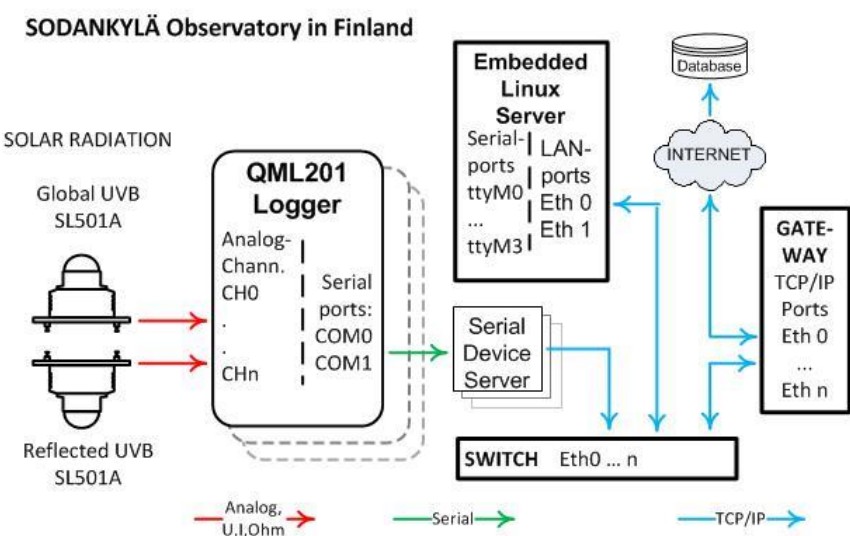

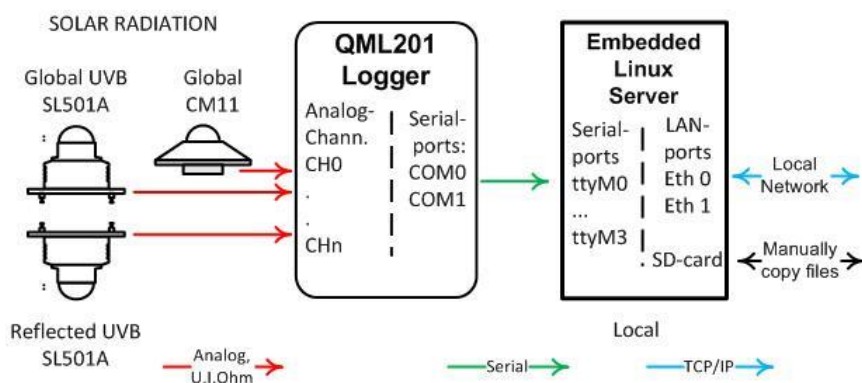



Figure 4. The block diagram of incoming and outgoing solar UV radiation measurements in
Sodankylä and Argentina using the SL-501A sensors.





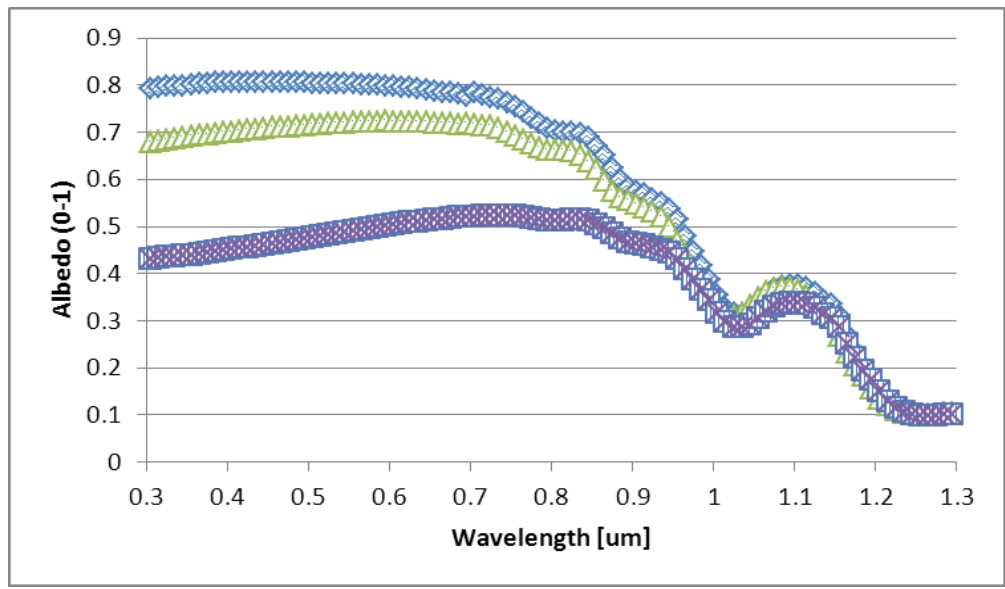



Figure 5. The effect of the light-absorbing BC on snow albedo is the biggest at the UV wavelengths.
The highest reflectivity is calculated for snow with no BC (upper curve, blue). If the amount of BC
is increased to 200 ppb and the mass absorption scaling factor is the default value 1.0, spectral
albedo is decreased, the more the smaller the wavelength (middle curve, green). Albedo is further
decreased, if for the same BC amount only the mass absorption scaling factor is increased (lower
curve, purple). Otherwise, the blue, green and purple curves are calculated using the same input
values. The figure demonstrates that it is both the BC concentration and the MAC values together
that affect the simulated spectral albedo values. The simulation was calculated using the SNICAR
model by Flanner et al. (2007).