# Peer review of "Bipolar long-term high temporal resolution broadband"

_Geoscientific Instrumentation, Methods and Data Systems, 2015_

## Referee Comment (RC1) · Anonymous Referee #1 · 16 Feb 2016

A) General comments

The manuscript is divided into two parts. First it presents the bipolar UV albedo monitoring station at Sodankylä and Marambio. It explains in full details the setup and the challenges operating these polar stations. The second part tries to give an extensive overview over albedo measurements, analysis and modeling of these kinds of data. The later chapters are defined as literature review by the authors.

The first part is a fundamental manuscript which could be used a master reference not only for further work related to the two measurements station but also to similar

projects.

The second part summarizes resent albedo studies. It references to already published work. Although the reader gets a nice overview over these works this section is strongly questionable. Does this literature review fit to the more technical section in the beginning of the paper? These summaries of many publication do not enhance the readability of the manuscript. Thus, on the one hand the reader gets a nice overview of the past studies but on the other hand – without reading the actual papers – it is difficult to grab the paragraphs of this literature review.

B) Specific scientific comments

The following comments should be taken into consideration to improve the quality of the manuscript:

1.) The setup in Marambio is fixed to a container which affects the snow deposition around it depending on the wind direction. How is this effect handled?

2.) Measuring height of 2m: If the snow accumulation around the setup is more than 1m than the distance between the snow surface and the instrument is below the recommended standard height. Does this never occur at both stations?

3.) It is unclear what the cleaning frequency of the entrance domes actually is.

4.) Is there any kind of ventilation and heating systems (VHS) around the devices? If not, how many measurements are affected by snow accumulated on the sensors?

5.) In line 191 the life time of sensors is discussed. Do the author mean the lifetime of the calibration or the device itself? Sensitivity changes of this sensor type are mostly affected by the lake of maintenance (old desiccant). This is independent on the light exposure and sensitivity changes happen most frequently in the storage rooms.

6.) The sensors of Marambio are calibrated in Finland which includes long distance transportation. Sensitivity changes are thus detected after the arrival in Finland. Air

travel can strongly affect the sensitivity of the sensor by the lower pressure present during the transport (humidity can enter the device). This can be tested during the calibration period at the calibration facility. Could the authors comment on this point?

C.) Presentation

First of all it should be considered to either move section 3 and 4 to a separate manuscript. Alternatively these to sections could by shortened to one section "Albedo literature review"with a few paragraphs listing the literature references. The emphasis of the paper as indicated by the title should be on the "measurement system" of the two stations. Currently the paper is divided approx. 60/40. Otherwise the manuscript is clearly structured. Minor modifications are recommended to improve the quality of the paper:

1.) Throughout the paper abbreviation are used either without declaration or they are multiple times declared. In addition, no common declaration style is used or late declarations are used. Consider using the standard style: "first use, first declaration". Examples without claiming to be complete: line 53 "RT", line 107 "SMN", line 217 "SZA", line 285 "VIS", ...,line 31 and 94 "IPY", line 44 and 59 "UV", line 29 and 77 WMO and (WMO), line 290 – first late declaration of "BC"

2.) The statement "first time" is used in line 24, 57, 69 and 332. As this is intrinsic for a novel manuscript it is not needed.

3.) Line 110: "The measurements" change to "The measurement devices" (or similar)

4.) Typo line 136: 2pi -> $2\pi$

5.) Paragraph line 154 to 160 is essentially a copy of the former (146-153). Both paragraphs could be merged together.

6.) Equation 2 (if the paragraph remains in the paper) should be written as: A=c*SZA, with A being the albedo decline, c the fit constant and the solar zenith angle SZA (if not previously defined!). In the Neumeyer data c found out to be -0.024 or -0.0024 (line

[Figure]

266 and 268)?

Figure 3: The location of the radiometers at the container is not visible.

Figure 5: Copy of Meinander, 2013. This meaning is only understandable in the context of the original publication.
[Figure]

---

## Referee Comment (RC2) · Anonymous Referee #2 · 19 Feb 2016

General comments

The manuscript of Meinander and others has two main aims: (1) to provide a technical description of UV radiation and albedo measurement stations in the Finnish Arctic (Sodankylä) and maritime Antarctic (Marambio), including an identification of potential errors; and (2) provide a literature review of existing publications (mainly scientific reports, which are presumably not widely available) at one site (Sodankylä). The first part provides a useful scientific reference for potential users of the data and for establishment of similar high quality UV measurement systems elsewhere (although the section

on measurement 'challenges' really needs to quantify error sources and provide methods to rectify them). The second part is rather ineffective consisting of a disparate collection of observations and somewhat speculative interpretation.

The paper could be shortened considerably by using more concise language and better structure to avoid the frequent repetition. There are also a couple of places were the language is convoluted and unclear, together with other basic problems of expression such as undefined acronyms.

Specific comments

1. The Marambio measurements appear to be made in a large container, although this isn't entirely clear. This setup needs a fuller explanation and an assessment of the effect of the container on the measurements through shadowing and impact on wind drift needs to be evaluated.

2. Section 2.5 identifies a number of potential error sources (mistermed as 'challenges'), but doesn't go any further than this. Without quantifying these errors and explaining how to remedy them this section isn't much use to potential users of the data.

3. Section 3 (the second part of the paper) is a rather disparate collection of observations, simplistic analysis and somewhat speculate interpretation which doesn't seem to have clear guiding objectives. In order to understand this section the underpinning data need to be displayed in graphs and tables and the measurement methods, for example how sampling for impurities was conducted, need to be explained. Some similar observations, e.g. daytime hysteresis, have also been made in broadband albedo measurements of snow, but there is no cross referencing to this literature to identify common explanations. Section 3 might be better presented as either a short section (1 or 2 paragraphs) just reporting the main findings, or else removed entirely and written up as a full review paper which could provide the required critical evaluation and cross-referencing.

Technical corrections

L18 'large amount of' is vague and can be removed

L24, and L57 and elsewhere in the paper 'for the first time' doesn't need to be stated as it should be obvious that a scientific paper is presenting something new

L24-L28 A very convoluted sentence which doesn't make grammatical sense. Rewrite as two or more sentences, and explain what the 'comparison' is between

L35 and L68-L70 'literature review' implies a critical evaluation of a broad sweep of published material which is not what you do. 'Summary' would be a better term

L36 explain these publications are mainly reports which are not widely available (if that is the case)

L53 RT is not defined

L57-60 Same comment as for L24-L28

L80 don't abbreviate 'figure' and be consistent in this throughout the paper

L93 insert 'a' before 'snow surface'

L103 'September or October' is vague, give the exact date

L173 This is not a good title for the section. 'Error sources and their treatment' would be a more informative title Section 2.5 This section could be written much more concisely

L181 '...regions, characterized by...'

L182-L183 'low solar elevation' vague, give the exact elevation

L185 'Due to frequent calm or low wind speeds...' would be a better way to start this line

L187 'really low' vague, give the exact elevations

L191 when does the sun appear exactly?

L192 again, give the solar elevation

L198 '…maintain clean and free…'; '…visited only once…'

L199-L202; wind appears 3 times as points ii), iv) and v). Isn't wind just a single hazard?

L200 what exactly is meant by 'uncomfortable' and is this really relevant to the scientific integrity of the data?

L205 Should this section be called 'Summary of results from previous studies using Sodankylä and Marambio data'

L206-L211 This overview of the paper is repetition and misplaced here. Integrate with section 1

L212-L215 repetition of section 2.3

L217 define 'SZA'

L234 define 'A'

Figure 4 only seems to show the UVB sensors and not the UVA sensor. Is there a reason for this?

Figure 5 please provide a legend. The caption should just explain what is shown in the figure, the description of methods used should be moved to the text of the paper.
* * *

---

## Referee Comment (RC3) · Anonymous Referee #3 · 8 Mar 2016

General comments

The manuscript by Meinander et al. describes a system for the measurement of incoming and outgoing solar broadband UV radiation at two polar sites, i.e. Sodankylä and Marambio. After a general introduction, the measurement sites, the working principles, the data collection system, the calibration and the biggest challenges in such measurements are illustrated. Starting from Sect. 3, the authors list the major findings already reported in their previous publications.

Although the paper addresses very relevant scientific questions, its declared focus is

"not to publish the existing data nor their scientific analysis", as the authors themselves acknowledge, and the manuscript does not present any new finding. Sections 1, 2.1.1, 2.2, 2.4, 2.5 and most of Sect. 3 are fundamentally quoted from Meinander et al., 2008; the rest of Sect. 3 and Sect. 4.1 are taken from Meinander et al., 2009, 2013 and 2014, respectively. The only new addition to the present manuscript is the description of the Antarctic Marambio Base and its instrument (which is, however, similar to the one employed in Sodankylä and already described by Meinander et al., 2008 in detail).

Honestly, I cannot find any reason to publish the manuscript in GI, unless relevant new findings are added to the text. In that case, Sects. 3-4 could be summarised to compose a sound introduction of a substantially new paper. The manuscript cannot even be considered, in my opinion, a complete literature review on snow UV albedo, since most of the cited references in the second part of the text only belong to the authors.

Specific comments

The authors list the main "challenges" of the bipolar UV radiation measurements, but they do not provide an adequate quantification of the resulting overall measurement uncertainty in UV albedo estimates. How large do they expect it to be? Do this kind of measurements still make sense even in presence of large uncertainties?

l. 70: "this is our first paper to consider the Sodankylä incoming irradiance as an independent data set". Please, explain how this is accomplished in the manuscript. Also, does this mean that no incoming UV irradiance measurements have been previously performed in Sodankylä?

Technical corrections

l. 23: "OC/BC": please, define acronyms when used for the first time

l. 28: "Global Atmosphere Watch (GAW)"

l. 29: "World Meteorological Organization (WMO)"

l. 31: "International Polar Year (IPY) 2007-2008"

l. 39-43: at least one citation is needed here

l. 53-54: define the "RT" acronym. This sentence is also quite confusing to the reader, since the quantity employed in radiative transfer models is effective albedo, not local albedo. Please, explain how the two quantities are related to each other

l. 126: add full stop at the end of the sentence

l. 135: the SL501 erythemal irradiance is not "calculated" as a spectral integral, since broadband instruments cannot measure a spectrum and convolve it to an action spectrum. Rather, the measurement "represents" the convolution of the solar irradiance spectrum to the spectral response function of the instrument

l. 144: why "spectral"?

l. 151: is it essential to always specify "linux-computer"?

l. 154-160: please, describe only the differences between both systems, do not repeat the common characteristics

l. 176: are the sensors only "temperature controlled" or also "temperature stabilized"?

l. 182: "some data": please, explain what kind of data needed to be excluded

l. 187: could you explain how you cope with the problem of tree shadows?

l. 212: "independent data of incoming and outgoing UV radiation": what do the authors mean by "independent"?

l. 266: how is "c" defined?

l. 308: "The Committee on Earth Observation Satellites (CEOS)"

l. 330-331: does this consideration apply to both "polar regions" (l. 328)?

---

## Author Comment (AC1) · 3 Jun 2016

**Authors' response to the review of the Anonymous Referee #1**

The Authors appreciate and fully agree to all of the constructive comments of the Referee #1. We also agree with the Referee that the comments of the Referee#1 will significantly improve the quality of the revised version of our manuscript. Our respond to each remark is given here below. (The language check will be made only to the final revised version of the manuscript.)

**A) General comments**

*Referee #1 comment:*
*"The manuscript is divided into two parts. First it presents the bipolar UV albedo monitoring station at Sodankylä and Marambio. It explains in full details the setup and the challenges operating these polar stations. The second part tries to give an extensive overview over albedo measurements, analysis and modeling of these kinds of data. The later chapters are defined as literature review by the authors. The first part is a fundamental manuscript which could be used a master reference not only for further work related to the two measurements station but also to similar projects.*

*The second part summarizes resent albedo studies. It references to already published work. Although the reader gets a nice overview over these works this section is strongly questionable. Does this literature review fit to the more technical section in the beginning of the paper? These summaries of many publication do not enhance the readability of the manuscript. Thus, on the one hand the reader gets a nice overview of the past studies but on the other hand – without reading the actual papers – it is difficult to grab the paragraphs of this literature review."*

*Authors' reply*
We thank the Referee#1 for this general comment and fully agree. We are also thankful for the sentence describing our paper as *"...master reference not only for further work related to the two measurements station but also to similar projects"*. This is exactly the motivation of the paper. We suggest to use this Referee's sentence in our revised manuscript, as indicated here below.

Our manuscript was originally planned to contain both the technical description, and thereafter a summary (which we called "review" in the original manuscript) showing what has been found in these measurement data so far. Based on the comments of all the three reviewers we suggest to use the word *"summary"* instead the word *"review"*. In addition we suggest to shorten the summary, and move it into the Introduction-section. These literature references are given for the benefit of any future data user. This we suggest to be more clearly stated in the revised version.

*Suggested changes in the manuscript:*

1) to add the following new sentences: *"The aim of this paper is to serve as a master reference not only for further work related to the two measurements stations, but also to similar projects by others".*
2) use *"summary"* instead of "review" when referring to our earlier results
3) shorten this literature summary from the original literature review (of our own previous results)
4) move the literature summary to the Introduction section.
* * *
**B) Specific scientific comments**

*Referee #1 comment 1:*
**"The following comments should be taken into consideration to improve the quality of the manuscript.**
**1.) The setup in Marambio is fixed to a container which affects the snow deposition around it depending on the wind direction. How is this effect handled?"**

*Authors' reply*
We fully agree that the Referee#1 detailed comments can be used to improve (and have improved) the quality of the manuscript. Our specific replies, point by point, are given here.

1)The Referee brings out an important question here. We fully agree that the wind can highly affect the snow deposition in Marambio. The effect is taken into consideration by taking photos of the snow surface on a weekly basis. Another solution could be a web camera showing the snow surface more frequently, and placing permanent snow height sticks (still avoiding to disturb the snow surface). Using the existing photographs we know the circumstances at the same moment as the photograph was taken. On the other hand, between the photos, snow surface conditions are unknown. Here clearly exists an improvement possibility for the future for Marambio work, or any snow albedo measurement setup. As a result of this Referee comment, an outdoor IP camera (pointing at the same view of the sensor), has actually been agreed to be installed besides the measurement. Marambio personnel will take the responsibility of installing the camera and programming it to run automatic a photo routine for a timelapsed, while FMI will provide the camera.

*Suggested changes in the manuscript (also shown in the revised ms prepared using "track changes")*
We suggest to add the following new sentence:

*"The setup in Marambio is fixed to a container which affects the snow deposition around it depending on the wind direction. This effect is taken into consideration by snow surface photographs taken on a weekly basis. An alternative solution for the problem is a web camera documenting the snow surface more frequently. This has recently been agreed to be installed besides the measurements. Currently, the Marambio SL501 measurement data are uploaded by automatic routine once per hour to the ftp site and also stored in FMI data base."*

*Referee #1 comment 2:*

*"2.) Measuring height of 2m: If the snow accumulation around the setup is more than 1m than the distance between the snow surface and the instrument is below the recommended standard height. Does this never occur at both stations?"*

*Authors' reply*

        In Sodankylä the maximum snow height measured during 1911-2016 has been measured to be 119 cm in winter 1999-2000 (ref, [http://ilmatieteenlaitos.fi/lumitilastot](http://ilmatieteenlaitos.fi/lumitilastot)). Hence, in Sodankylä it is possible to happen. Snow albedo changes according to snow properties, of which the grain size the most critical. On the other hand, albedo is a quantity that also changes according to the measurement height. Therefore, a change in the snow depth also has the potential to affect the measured albedo. It is however currently practically impossible to change manually the albedo measurement height often enough to keep a fixed distance between the downward looking sensor and the snow surface, without disturbing the albedo field. Automatical height change for the albedo sensors would be ideal, but not available. Therefore, we handle this by: automatically measuring the snow height closeby the Sodankylä albedo measurement field, and albedo is measured in a fixed position.

        In Marambio, this amount of snow does not occur. In the surrounding area of the Marambio shelter, after the event that heavy snow and/or blizzard, some 10- 30 cm of snow during winter or spring time could collected.

***Suggested changes in the manuscript (also shown in the revised ms prepared using "track changes")***

We suggest to add the following text:

        *"The WMO defined albedo measurement height is 1-2m. If the snow accumulation around the setup is more than 1m, the distance between the snow surface and the instrument is below the recommended standard height. This can sometimes happen in Sodankylä, where the maximum snow height ever measured (1911-2016) is 119 cm on 6 April 2000 (ref, [http://ilmatieteenlaitos.fi/lumitilastot](http://ilmatieteenlaitos.fi/lumitilastot)). Albedo is a quantity that changes according to the measurement height, although snow grain size is the most critical parameter to determine snow albedo. Therefore a natural change in the snow depth also has the potential to affect the measured albedo. As a result, we measure automatically the snow height closeby the Sodankylä albedo measurement field, while albedo is measured in a fixed height. Alternatively, the measurement height could, if possible in practice, automatically changed to keep a fixed distance between the downward looking sensor and the snow surface. A manual change in adjusting measurement height to achieve a fixed distance between snow surface and the sensor can be considered if it is possible without disturbing the measurement or destroying the snow surface.In Marambio, this much snow is not an issue"*

*Referee #1 comment 3:*
*"3.) It is unclear what the cleaning frequency of the entrance domes actually is."*

*Authors' reply*

First we need to consider that any visit to manually clean the domes disturbs the measurement and the snow surface. On the other hand, if the dome is dirty, we can't rely on the data. Manual cleaning is therefore done when needed but as seldom as possible to avoid disturbing the measurements. The automatical blowing and sensor temperature regulating systems (a Peltier

element to keep the sensor temperature at 25 deg Celsius; this temperature is measured and reported together with the data at 1 min intervals) partially help to keep the domes cleaner.

In the FMI Sodankylä Arctic Research Center, all the radiation sensors are cleaned always when needed, which means minimum of once a week during snow time. The best estimate is 5 times/month. Sodankylä  sensor domes have blowing systems which keep them free of falling snow, and also the blowing dries water droplets away from the domes.

In Marambio, the domes are cleaned once per week. This depends of the weather conditions, which can extend the length of time, that can elapse in some cases between two visits to the shelter (up to 10 days). In fact, the heating system can also clean the domes a bit in certain conditions, but the domes usually gets dirty when it snowing and the permafrost is driven with the wind. The heating system is not so powerful to remove this particles. The upper dome usually is more dirty than the other one. Sometimes there are a little snow (or piece of ice) in the upper dome accumulate. The position of this piece of ice (or snow) cover a small part of the North-East dome sector.

Hence, all the sensors both the Marambio and Sodankylä have blowers on the domes and the sensors have Peltier elements for temperature regulation for 25 deg Celsius.

***Suggested changes in the manuscript (also shown in the revised ms prepared using "track changes")***
We suggest to add the following new sentence:
*"The temporal frequency of the manual cleaning of the entrance domes balances between the facts that manual cleaning disturbs the measurement and changes the surface snow (in the North direction) and that there is a need to clean  the domes from dirt to gain as reliable data as possible. In Sodankylä and Marambio the domes are manually cleaned once per week, or when needed."*

***Referee #1 comment 4***
***"4.) Is there any kind of ventilation and heating systems (VHS) around the devices? If not, how many measurements are affected by snow accumulated on the sensors?"***

***Authors' reply***
Yes, both in Marambio and Sodankylä the sensors are heated (temperature regulated using a Peltier element to set the sensor temperature to 25 deg Celsius).

***Suggested changes in the manuscript (also shown in the revised ms prepared using "track changes")***

Instead of saying "temperature controlled", we will we use  *"temperature regulated"*.

***Referee #1 comment 5***
***5.) In line 191 the life time of sensors is discussed. Do the author mean the lifetime of the calibration or the device itself? Sensitivity changes of this sensor type are mostly affected by the lake of maintenance (old desiccant). This is independent on the light exposure and sensitivity changes happen most frequently in the storage rooms.***

***Authors' reply***

In line 191 we refer to the lifetime of the sensors. We fully agree that sensitivity changes occur even when storing the sensors.

***Suggested changes in the manuscript (also shown in the revised ms prepared using "track changes")***

The revised version to clarify to what the term "lifetime" refers:

*"To prolong the lifetime of the sensor devices, they are not kept outside when the Sun is at lowest. Sensitivity changes are expected to occur also independent on the light exposure. Therefore the responses of the sensors need to be measured on a regular basis. In our case, It means before and after each measurement season of less than 6 months in Sodankylä and one year in Marambio (up to 2 years)."*

*Referee #1 comment 6*
***6.) The sensors of Marambio are calibrated in Finland which includes long distance transportation. Sensitivity changes are thus detected after the arrival in Finland. Air travel can strongly affect the sensitivity of the sensor by the lower pressure present during the transport (humidity can enter the device). This can be tested during the calibration period at the calibration facility. Could the authors comment on this point?***

*Authors' reply*
The Referee makes an important note on the air travel affecting the sensors, which we had not included in the original manuscript . For Sodankylä, air travel does not affect, since the sensors are transported by car. For Marambio, the sensors air travel from Finland to Antarctica. After calibration, the calibrated sensors are transported to Marambio

***Suggested changes in the manuscript (also shown in the revised ms prepared using "track changes")***

*To be added: "The sensors of Marambio are calibrated in Finland. The calibrated sensors are then transported to Antarctica in airplanes with long distance transportation. Air travel can in principle strongly affect the sensitivity of a sensor by the lower pressure present during the transport or changes in the humidity entering the devices. Any sensitivity changes are detected and corrected during the calibrations in Finland. As the SL501 sensors are pressurized with nitrogen, the air pressure changes during air travel are not assumed to affect the sensors, and we have not detected any indications of that. For Sodankylä sensors, air travel does not affect, since the sensors are transported by car.*
* * *
**C.) Presentation**
*Referee #1 comment:*
***"First of all it should be considered to either move section 3 and 4 to a separate manuscript. Alternatively these to sections could by shortened to one section "Albedo literature review" with a few paragraphs listing the literature references. The emphasis of the paper as indicated by the title should be on the "measurement system" of the***

*two stations. Currently the paper is divided approx. 60/40. Otherwise the manuscript is clearly structured."*

*Authors' reply*
We fully agree to the Referee comment that the emphasis of this paper should be, "*as indicated by the title, on the "measurement system" of the two stations*". We also agree to the alternative solution suggested by the Referee for better re-organizing the manuscript: i) we agree to the Referee suggestion that sections 3 and 4 could be moved to a separate manuscript. ii) Also we agree that alternatively a shortened section on albedo literature review could be a good idea. To have the focus on the main task of the paper (measurement system) we suggest to shorten the review, and name it a summary, and give the references in the introduction section.

**Suggested changes in the manuscript (also shown in the revised ms prepared using "track changes")**
We refer here to our reply in A).

*Referee #1 comment:* *"Minor modifications are recommended to improve the quality of the paper:"*

*1.) Throughout the paper abbreviation are used either without declaration or they are multiple times declared. In addition, no common declaration style is used or late declarationsare used. Consider using the standard style: "first use, first declaration".*
*Examples without claiming to be complete: line 53 "RT", line 107 "SMN", line 217 "SZA", line 285 "VIS", ...,line 31 and 94 "IPY", line 44 and 59 "UV", line 29 and 77 WMO and (WMO), line 290 – first late declaration of "BC*
- *Authors' reply: We agree that these need to be corrected in the way suggested by the Referee.*

*2.) The statement "first time" is used in line 24, 57, 69 and 332. As this is intrinsic for a novel manuscript it is not needed.*
- *Authors' reply: We agree and remove multiple "first times".*

*3.) Line 110: "The measurements" change to "The measurement devices" (or similar)*
- *Authors' reply: We agree to change "the measurements" to "the measurement devices".*

*4.) Typo line 136: 2pi -> 2_*
- *Authors' reply: Agree, this needs to be corrected.*

*5.) Paragraph line 154 to 160 is essentially a copy of the former (146-153). Both paragraphs could be merged together.*
- *Authors' reply: Agree, these paragraphs need to be merged together.*

*6.) Equation 2 (if the paragraph remains in the paper) should be written as: A=c\*SZA, with A being the albedo decline, c the fit constant and the solar zenith angle SZA (if not previously defined!). In the Neumeyer data c found out to be -0.024 or -0.0024 (line 266 and 268)?*
- *Authors' reply: Agree on the Eq 2, where A is for albedo decline.*

***Figure 3: The location of the radiometers at the container is not visible.***

- ***Authors' reply:*** *Agree, the photo only shows the surroundings of the containor prior to adding the measurement devices.We will add a new figure (below) showing the pole where the sensors will be (are) attached.*

[Figure]

.

***Figure 5: Copy of Meinander, 2013. This meaning is only understandable in the context of the original publication.***

- ***Authors' reply:*** *Agree that the parameter values need to be given in the revised paper in addition to the literature reference.*

---

## Author Comment (AC2) · 21 Jun 2016

**Authors' response to the review of the Anonymous Referee #2**

The Authors are thankful for the valuable review comments given by the Referee #2. The comments of the Referee#2 are helpful to improve the quality of the revised version of our manuscript. Our respond to each remark of the Referee#2 is given here below. (The language check will be made only to the final revised version of the manuscript.)

It can be noted that our manuscript was originally planned to contain both the technical description, and thereafter a summary (which we called "review" in the original manuscript) showing what has been found in these measurement data so far. Based on the comments of all the three reviewers we suggest use the word *"summary"* instead of the word *"review"*. In addition, we suggest shorten the contents of this summary to 1-2 chapters, as suggested by Referee#2, and move it into the Introduction-section. The literature references on our earlier work are given for the benefit of any future data user. This fact we suggest to be more clearly stated in the revised version, too.

   *A) General comments*

*Referee #2 comment:*
**"The manuscript of Meinander and others has two main aims: (1) to provide a technical description of UV radiation and albedo measurement stations in the Finnish Arctic (Sodankylä) and maritime Antarctic (Marambio), including an identification of potential errors; and (2) provide a literature review of existing publications (mainly scientific reports, which are presumably not widely available) at one site (Sodankylä). The first part provides a useful scientific reference for potential users of the data and for establishment of similar high quality UV measurement systems elsewhere (although the section on measurement 'challenges' really needs to quantify error sources and provide methods to rectify them). The second part is rather ineffective consisting of a disparate collection of observations and somewhat speculative interpretation. The paper could be shortened considerably by using more concise language and better structure to avoid the frequent repetition. There are also a couple of places were the language is convoluted and unclear, together with other basic problems of expression such as undefined acronyms."**

**Authors' reply**
         We agree totally to the contents of the general comment given by the Referee#2. We are most pleased with the Referee#2 statement saying that "***The first part provides a useful scientific reference for potential users of the data and for establishment of similar high quality UV measurement systems elsewhere".*** We also agree that it would be most useful to quantify the error sources (**Referee#2 comment** *"the section on measurement 'challenges' really needs to quantify error sources and provide methods to rectify them").* However, to properly include and quantify all the error sources can, in our opinion, easily be the subject of an entire paper itself. Therefore, we suggest to include here in the revised version of this manuscript some of the first quantifications of error sources, as specified in the detailed replies due toReferee#2 specific comments.

We also agree to the Referee#2 general comment on the second part, and suggest to replace review by a shortened summary, and move this summary in the Introduction-section. The literature references on our earlier work are given for the benefit of any future data user. This fact we suggest to be more clearly stated in the revised version, too. The language check will be made to the final revised version of the manuscript.
* * *
**B) Specific comments**

**Referee #2 comment 1**
***1. The Marambio measurements appear to be made in a large container, although this isn't entirely clear. This setup needs a fuller explanation and an assessment of the effect of the container on the measurements through shadowing and impact on wind drift needs to be evaluated.***

**Authors' reply**
We agree that a clarification on the measurement place is needed. A new photograph (below) is suggested to be inserted in the revised manuscript. The photo shows the horizontal pole where the downward sensor will be (is) attached (the pole will be/has been put as far from the container as possible; in this figure the pole is attached from the middle) .

We agree with the referee that the effects of shadowing and wind drift are important to be included in the considerations of the revised manuscript. We agree that more effort needs to put in describing both of these effects, as will be explained here below. However, we also state that both of these effects could be a subject of an entire paper, and such papers have been published, too.

Thus, the shadowing effects due to the container could, in our opinion, be an interesting subject of its own independent paper. Earlier, we have presented calculations for a similar albedo edge effect case in Meinander and Räisänen (2010) http://www.atmos-chem-phys-discuss.net/10/C11474/2010/acpd-10-C11474-2010-supplement.pdf). Therein, in case of albedo measurements on snow patch with size 80 m North-South, and 50 m East-West), and at 3 m distance in one direction (North), assuming the true snow albedo were $\alpha$(snow) = 0.7 and the environmental albedo $\alpha$ (env) = 0.1, the measured albedo would be 0.629, indicating an error of -0.071 in the measured albedo.

In Marambio, the container was placed in the middle of an open large field with >100 m by >100 m and sensor was placed toward the Sun (North), and the distance to one direction (South), i.e., the container, was appr. 2-3 m. The Marambio measurement height was appr. the same (2 - 2.5 m) as for Meinander and Räisänen (2010) case. Therefore, considering these distances, the error due to the container in the albedo measurements are not to be expected to be bigger than the error caused in the case of Meinander and Räisänen (2010).

However, as the snow surface is in practice not flat in Marambio, due to wind drift, as brought up by the Referee#2, some of the assumptions presented in Meinander and Räisänen (2010) calculations are violated in case of Marambio. The effects of wind drift on albedo in Marambio can, however, be identified using the photographs taken on a weekly basis for the albedo measurements.

As an outcome, we suggest to include in the revised manuscript a first estimation to the shadowing effect of the container, in case of flat snow surface, based on the case by Meinander and Räisänen (2010). We also suggest including in the revised manuscript a more detailed description of the photographs and their availability for the albedo data user. For the incoming solar irradiance data, the shadowing and wind drift effects can only affect via multiple reflection from the environment reaching the upward looking sensor. Changes in these effects from the irradiance point of view we state to be insignificant for the usage of the irradiance data.

[Figure]

**Referee #2 comment 2**
*2. Section 2.5 identifies a number of potential error sources (mistermed as 'challenges'), but doesn't go any further than this. Without quantifying these errors and explaining how to remedy them this section isn't much use to potential users of the data.*

**Authors' reply**

We agree with the Referee#2that all the error sources are finally needed to be identified and quantified. How ever, as stated in our reply to the Referee#2 Comment 1, the effect of these errors could be subject of their own papers. Therefore, we have also given the name "challenges" to these error sources, that are only identified, not quantified. We suggest to state in the revised version of our manuscript that "This paper identifies the known error sources affecting these data, and gives the first estimates of the range of these errors, as error sources such as shadowing or wind drift effect could be a subject of their own papers." These error estimates will be based on our own calculations (e.g., Meinander and Räisänen 2010), or literature references, such as WMO (1996), Hulsen and Grobner (2007), given in Meinander et al. (2008):

*"Here, use was made of erythemal UV albedo measurements*
*by broadband SL501 radiometers with similar spectral re-*
*sponses, thus resulting in errors of less than 1% due to dif-*
*ferences in the sensors (WMO, 1996). According to Hulsen*
*and Grobner (2007), the typical total uncertainty for SL501*

*instruments is from 1.7 to 4.3 %. "*

In addition, we suggest to add in the revised manuscript the following Equation of Briegleb et al. (1986) to explain the U-shape of the detected albedo:

$$R(\mu) = R_0 \frac{(1+d)}{1+2d\mu}$$

where $\mu$ is the cosine of the SZA, and $R_0$ is the reflectivity for $\mu = 0.5$ as given in their Table 2, and $d$ is an empirical parameter.

Reference:

Briegleb BP, Minnis P, Ramanathan V, Harrison E. Comparison of Regional Clear Sky Albedos Inferred from satellite Obervations and Model Computations. Journal of Climate and Applied Meteorology, 25, 214-, 1986.

This Equation we think can be of use for the albedo data user to understand the SZA dependency of the albedo data.

*Referee #2 comment 3.*
*3. Section 3 (the second part of the paper) is a rather disparate collection of observations, simplistic analysis and somewhat speculate interpretation which doesn't seem to have clear guiding objectives. In order to understand this section the underpinning data need to be displayed in graphs and tables and the measurement methods, for example how sampling for impurities was conducted, need to be explained. Some similar observations, e.g. daytime hysteresis, have also been made in broadband albedo measurements of snow, but there is no cross referencing to this literature to identify common explanations. Section 3 might be better presented as either a short section (1 or 2 paragraphs) just reporting the main findings, or else removed entirely and written up as a full review paper which could provide the required critical evaluation and cross-referencing.*

**Authors' reply**
We totally agree and will present the second part of the manuscript shortened to 1-2 paragraphs, and as a part of the Introduction.

**c) Technical corrections**

**Authors' reply**
The authors are grateful of all these detailed technical corrections notified by the Referee#2. Changes will be made accordingly.

*L18 'large amount of' is vague and can be removed:* agree

*L24, and L57 and elsewhere in the paper 'for the first time' doesn't need to be stated as it should be obvious that a scientific paper is presenting something new:* agree

***L24-L28 A very convoluted sentence which doesn't make grammatical sense. Rewrite
as two or more sentences, and explain what the 'comparison' is between:*** agree and will explain

***L35 and L68-L70 'literature review' implies a critical evaluation of a broad sweep of
published material which is not what you do. 'Summary' would be a better term:*** agree, summary
will be used

***L36 explain these publications are mainly reports which are not widely available (if that
is the case):*** agree, will be explained but also this will  be shortened

***L53 RT is not defined:*** agree, will define

***L57-60 Same comment as for L24-L28:*** agree

***L80 don't abbreviate 'figure' and be consistent in this throughout the paper:*** agree

***L93 insert 'a' before 'snow surface':*** agree

***L103 'September or October' is vague, give the exact date:*** agree

***L173 This is not a good title for the section. 'Error sources and their treatment' would be
a more informative title Section 2.5 This section could be written much more concisely:*** agree

***L181 'regions, characterized by:*** agree

***L182-L183 'low solar elevation' vague, give the exact elevation:*** agree

***L185 'Due to frequent calm or low wind speeds' would be a better way to start this
line L187 'really low' vague, give the exact elevations:*** agree, will give

***L191 when does the sun appear exactly?*** will give the exact when

***L192 again, give the solar elevation:*** agree

***L198 'maintain clean and free'; 'visited only once':*** agree

***L199-L202; wind appears 3 times as points ii), iv) and v). Isn't wind just a single
hazard?*** in the revised version we will give a more detailed description of the wind velocities in
Marambio

***L200 what exactly is meant by 'uncomfortable' and is this really relevant to the scientific
integrity of the data?*** agree, does not concern the data user, will be removed.

***L205 Should this section be called 'Summary of results from previous studies using
Sodankylä and Marambio data':*** agree, will use the summary word, and will shorten the contents
of the summary.

***L206-L211 This overview of the paper is repetition and misplaced here. Integrate with***

*section 1:* agree

*L212-L215 repetition of section 2.3:* agree, we thank the reviwer for the careful reading of the ms.

*L217 define 'SZA':* agree, will define

*L234 define 'A':* agree, will define

*Figure 4 only seems to show the UVB sensors and not the UVA sensor. Is there a reason for this?* We have only the UVB sensors.

*Figure 5 please provide a legend. The caption should just explain what is shown in the figure, the description of methods used should be moved to the text of the paper:* agree
* * *
Helsinki, 21 June 2016

The revised version of the manuscript will be provided after all the comments of the all three Referees have been replied to, and all the changes due to all these comments have been implemented.

Prior to our reply to Referee#2 here, we have given our reply to Referee#1. Next, we will reply to Referee#3.

Sincerely,
Outi Meinander, on the behalf of the co-authors

---

## Author Comment (AC3) · 22 Jun 2016

**Authors' response to the review of the Anonymous Referee #3**

We thank Referee#3 for giving critical review comments on our GID-manuscript of *"Bipolar long-term high temporal resolution broadband measurement system for incoming and outgoing solar UV radiation, and snow UV albedo, at Sodankylä (67°N) and Marambio (64°S)"* Our respond to each remark of the Referee#3 is given here below.

First, due to the critical comments of Referee#3, we will in our reply (more detailed here below) suggest some new unpublished quantitative results that we consider relevant for any data user. We will give some example figures to demonstrate some of the options of such results.

Secondly, we'd like to bring out that all the three reviewers gave the same comment on the second section. The common comment was to shorten and summarize the contents of the second section, and rather use it as an introductory summary. We therefore suggest to follow this advice, and to shorten the contents of the summary (named as summary, instead of "review" used in the submitted manuscript) to 1-2 chapters, and to include this in the Introduction-section. The literature references of our earlier work are given for the benefit of any future data user. This fact we suggest also to be more clearly stated in the suggested revised version of the manuscript. In addition, we suggest to include a more detailed theoretical background in the Introduction, with key equations to explain the data, and also including some quantitative error and uncertainty estimates of these data. Some of these will be suggested here, and some of these are explained in our replies to Referee#3 comments (below later).

For the U-shape of the albedo signal, we suggest to add the following Equation of Briegleb et al. (1986) to explain the U-shape of the detected albedo:

$$R(\mu) = R_0 \frac{(1+d)}{1+2d\mu}$$

where $\mu$ is the cosine of the SZA, and $R_0$ is the reflectivity for $\mu = 0.5$ as given in their Table 2, and $d$ is an empirical parameter.

Reference:

Briegleb BP, Minnis P, Ramanathan V, Harrison E. Comparison of Regional Clear Sky Albedos Inferred from satellite Obervations and Model Computations. Journal of Climate and Applied Meteorology, 25, 214-, 1986.

This Equation we think can be of use for the albedo data user to understand the SZA dependency of the albedo data.

Also, a new photograph (below) is suggested to be inserted in the revised manuscript. The photo shows the horizontal pole where the downward sensor will be/is attached.

[Figure]

In addition, related to measurement errors and uncertainties, we suggest to add 1-2 new edited sentences referring to Meinander et al. (2009) where we said (based on calculations presented by one of the co-authors, prof. Seckmeyer, Germany): *" the measured angular responses of the two Arctic SL-501 biometers were used to quantify uncertainties due to cosine error. Integrating incoming radiances over the whole hemisphere, and assuming isotropic distribution of the diffuse scattered light, we calculated an error of the incoming scattered light contribution of  0.5 % and 3.2 % for the up-welling and down-welling sensor, respectively."* Related to the measurement errors and uncertainties, we also suggested in our replies to Referee#1 and Referee#2  to include other new text in the suggested revised version of the manuscript. We will not repeat these here, as they are available for Referee#3 at http://www.geosci-instrum-method-data-syst-discuss.net/gi-2015-31/.

Finally, we'd also like to bring out the fact that our paper is aimed at GI, not elsewhere. The aims and scopes of the paper are defined as follows (http://www.geoscientific-instrumentation-methods-and-data-systems.net/about/aims_and_scope.html):

"Geoscientific Instrumentation, Methods and Data Systems (GI) is an open-access interdisciplinary electronic journal for swift publication of original articles and short communications in the area of geoscientific instruments. It covers three main areas: (i) atmospheric and geospace sciences, (ii) earth science, and (iii) ocean science. A unique feature of the journal is the emphasis on synergy between science and technology that facilitates advances in GI. These advances include but are not limited to the following:

- concepts, design, and description of instrumentation and data systems;
- retrieval techniques of scientific products from measurements;
- calibration and data quality assessment;
- uncertainty in measurements;
- newly developed and planned research platforms and community instrumentation capabilities;
- major national and international field campaigns and observational research programs;
- new observational strategies to address societal needs in areas such as monitoring climate change and preventing natural disasters;
- networking of instruments for enhancing high temporal and spatial resolution of observations.

GI has an innovative two-stage publication process involving the scientific discussion forum Geoscientific Instrumentation, Methods and Data Systems Discussions (GID), which has been designed to do the following:

- foster scientific discussion;
- maximize the effectiveness and transparency of scientific quality assurance;
- enable rapid publication;
- make scientific publications freely accessible."

We argue that our paper includes from these the following:
- **concepts, design and description of instrumentation and data systems** (here: bipolar SL-501 UV radiation measurements of incoming and outgoing solar radiation),
- **some of the calibration and data quality assessment and uncertainty** (here: we refer to the listed "challenges" of the submitted version, now suggested to be changed to more quantitative presentation of calibration, data quality and uncertainty, as presented in our reply to Referee#2 (ref. our reply to Referee#2 at http://www.geosci-instrum-method-data-syst-discuss.net/gi-2015-31/),
- **observational research programs** (here: WMO GAW Marambio and GAW Sodankylä; and Antarctic research under the FINNARP program),
* * *
**A) *General comments**

**Referee#3:**
*"The manuscript by Meinander et al. describes a system for the measurement of incoming and outgoing solar broadband UV radiation at two polar sites, i.e. Sodankylä and Marambio. After a general introduction, the measurement sites, the working principles, the data collection system, the calibration and the biggest challenges in such measurements are illustrated. Starting from Sect. 3, the authors list the major findings already reported in their previous publications. Although the paper addresses very relevant scientific questions, its declared focus is "not to publish the existing data nor their scientific analysis", as the authors themselves acknowledge, and the manuscript does not present any new finding. "*

**Author's reply:**
   To start with, we thank Referee#2 for saying that the paper addresses very relevant scientific questions.
   We agree with Referee#3 that it was our statement in the submitted manuscript that our focus was *"not to publish the existing data nor their scientific analysis".* We agree that our submitted manuscript did not contain new data nor their scientific analysis.
   To publish the incoming and outgoing measurement data with QA/QC and data analysis would require work similar to described in two ACP papers of Meinander et al. 2008 (http://atmos-chem-phys.org/8/6551/2008/acp-8-6551-2008.pdf) and Meinander et al. 2013 (http://www.atmos-chem-phys.net/13/3793/2013/acp-13-3793-2013.pdf), which both, in our opinion, show that there is a need for separate paper for publishing the data sets. (Data plots of raw data of incoming and outgoing solar radiation we suggest not to be published within this manuscript.)
   However, as will be explained more detailed here later, we have some new unpublished measurement data, e.g., on spectral and cosine responses of the sensors, which in our opinion could be useful for any data user, and could be published as part of this manuscript.
   Referee#3 then continues that " *the manuscript does not present any new finding".* To this we on the one hand partly agree, but on the other hand also partly disagree. First, we need to consider what is meant by "new finding". If finding refers to presenting new data and their analysis, it is true

that our submitted manuscript did not contain any new finding. This is because the focus of this paper is to present the measurement systems, and our experiences using them, for the benefit of a future data user. This is because we have decided to give the data out in data basis outside our own institutes. However, for the revised manuscript we will here below suggest new findings and quantitative data which are relevant to the paper and to data users, as well.

As background, we'd also like to bring out that the preparation of this manuscript was the first time that the people who had worked for these bipolar measurements in Finland and in Argentina, including Marambio and Sodankylä station technical personnel, gathered their experiences and work together. This certainly is of value, and this to take place is actually thanks to the existence of the journal GI. Hence, our paper aims for GI, not elsewhere. Keeping these aims and scopes of GI in mind, we argue that our paper includes from these the following (as said in the very beinning of our reply): **concepts, design and description of instrumentation and data systems** (here: bipolar SL-501 UV radiation measurements of incoming and outgoing solar radiation); **some of the calibration and data quality assessment and uncertainty** (here: we refer to the listed "challenges" of the submitted version, now suggested to be changed to more quantitative presentation of calibration, data quality and uncertainty, as presented in our reply to Referee#2 (ref. our reply to Referee#2 at http://www.geosci-instrum-method-data-syst-discuss.net/gi-2015-31/); **observational research programs** (here: WMO GAW Marambio and GAW Sodankylä; and Antarctic research under the FINNARP program). Therefore, we suggest more emphasis on these aims in the suggested revised manuscript. Additionally, we suggest new data to be included, as demonstrated in the figures and text here below.

As we stated already in the beginning of our reply to Referee#3, our own earlier publications (called as review in our submitted version) will be shortened and can be considered as a summary of a previous work, aimed to benefit any data user in the future. Therefore, such a summary can be of value, too.

**Referee#3 (continued):**

*"Sections 1, 2.1.1, 2.2, 2.4, 2.5 and most of Sect. 3 are fundamentally quoted from Meinander et al., 2008; the rest of Sect. 3 and Sect. 4.1 are taken from Meinander et al., 2009, 2013 and 2014, respectively. "*

**Author's reply:**

We agree this was the case in the submitted version, but this is not the contents of the revised version. In the revised version these are shortened, summarized and included in the Introduction, as suggested by Referee#3.

**Referee#3 (continued):**

*The only new addition to the present manuscript is the description of the Antarctic Marambio Base and its instrument (which is, however, similar to the one employed in Sodankylä and already described by Meinander et al., 2008 in detail).*

**Author's reply:**

We thank the referee for this point and agree that Marambio measurement description has not been published previously.

**Referee#3 (continued):**

*Honestly, I cannot find any reason to publish the manuscript in GI, unless relevant new findings are added to the text. In that case, Sects. 3-4 could be summarised to compose a sound introduction of a substantially new paper.*

**Author's reply:**

We thank Referee#3 for this very critical comment. We would suggest to include the summarized sections 3-4 in the Introduction as suggested by Referee#3 here.

We have seriously considered the Referee#3 comment saying "… ***unless relevant new findings are added to the text"***. As described below, including our reasoning, we'd like to suggest to include some new unpublished measurement data, related to the sensors and the measurement environemtn, in the revised manuscript, due to the comment of Referee#3.

Such relevant new findings that could be considered to be presented in the context of describing the measurement systems, and without presenting the data or its analysis (as it is not the scope of this paper, where we aim to describe the measurements could consist of, in our opinion, for example:

1) the previously unpublished measurement results of the cosine and spectral responses of all the sensors used for the measurements since IPY 2007/2008 for Sodankylä, and since 2013 for Marambio. Since 2013, we have always 4 sensors in use at a time. Previously, only one such result in one figure has been published in Meinander et al. (2008) for the 2 sensors used therein (Fig 1. of http://www.atmos-chem-phys.net/8/6551/2008/acp-8-6551-2008.pdf), also shown here below).Such data could either
   a) e.g, consist of all the cases in one figure for cosine responses and other for spectral responses, to show the minimum and maximum changes and the average values; or alternatively
   b) e.g., show more detailed indicating these curves for each used sensor together with a table identifying which sensor was used, where and when.

[Figure]

**Fig. 1.** Spectral (above) and cosine (below) responses of the SL501 sensors. Spectral responses are in logaritmic scale showing the maximal differences. The responses of the upward and downward sensors need to be considered when albedo results are interpreted.

The Figure above is adapted from Meinander et al, ACO, 2008. This kind of measurement results on spectral and cosine responses of the SL501 sensors used for Marambio and Sodankylä incoming and outgoing measurements have not been published otherwise, except this one figure for one pair of sensors used in Sodankylä for those data published therein. In Marambio and Sodankylä we have continuously 2 pairs of sensors in use. As a result of the critical comment of Referee#3, we suggest to include and publish in the revised version all the results of the measurements on spectral and cosine responses. These data are new previously unpublished quantitative results that can be of use for a data user.

alternatively/additionally

2) the effect of the tree cut in Sodankylä is evident in the data, and for this purpose we could give out some data showing that for, e.g., the pyranometer maximum values changed from appr. 0.7 to close to 1, after the tree cut. In W/m$^2$ data the change is not as pronounced as when looking at the albedo values.

[Figure]

[Figure]

The two figures above clearly demonstrate the change in the level of the measured albedo after the tree cut in Sodankylä. The albedo field is free of trees. In {W/m$^{2]}$ of incoming and outgoing solar radiation, the shadowing effect is not that pronounced (above), but in albedo data (0-1) the effect is clear. Although the raw data without further analysis and QA/QC procedures is not of value of publishing otherwise, we suggest that for the purpose of showing the shadowing effects of the environment in the boreal zone, this kind of figure can be of use for the data user.

> *The manuscript cannot even be considered, in my opinion, a complete literature review on snow UV albedo, since most of the cited references in the second part of the text only belong to the authors.*

**Author's reply:**

> We agree totally, and the word review is not used to describe the shortened summary planned to be included in the suggested revised version of the manuscript.
* * *
**c) Specific comments**

*Referee#3*

*The authors list the main "challenges" of the bipolar UV radiation measurements, but they do not provide an adequate quantification of the resulting overall measurement uncertainty in UV albedo estimates. How large do they expect it to be? Do this kind of measurements still make sense even in presence of large uncertainties?*

**Author's reply:**

We thank Referee#3 for this critical and very relevant comment and question. We refer to our paper of Meinander et al. (2008) saying that "Here, use was made of erythemal UV albedo measurements by broadband SL501 radiometers with similar spectral responses, thus resulting in errors of less than 1% due to differences in the sensors (WMO, 1996). According to Hulsen and Grobner (2007), the typical total uncertainty for SL501 instruments is from 1.7 to 4.3 %. "

We also argue that a transparent presentation on all the uncertainty and error sources of these data are needed in order the data user to use the data successfully used. Also, in the measurements such uncertainties and errors are expected, but not always brought up even in a qualitative way. We presented in our submitted manuscript our sincere overall understanding of all the factors affecting these data, and argue that after knowing all these error sources a successful scientific data usage is possible. In fact, some more discussion on errors and uncertainties is to be included in the suggested revised version as outcome of the comments given by all the three Referees (ref. http://www.geosci-instrum-method-data-syst-discuss.net/gi-2015-31/). Yet, the aim of this paper is not to present more detailed calculations of all the error sources. Many of these would be a subject of its own paper (such references are found in literature, and would be possible to do for these measurements, too).

*Referee#3:*

*l. 70: "this is our first paper to consider the Sodankylä incoming irradiance as an independent data set". Please, explain how this is accomplished in the manuscript. Also, does this mean that no incoming UV irradiance measurements have been previously performed in Sodankylä?*

**Author's reply:**

We need to clarify this in our suggested revised version of the manuscript. We refer here to the fact that our previous papers on this measurement setup for Sodankylä UV albedo on the operational albedo field have presented only results on UV albedo, although the incoming irradiance data could have been used alone, too. In turn, UV irradiance measurements with various instrumentations have been performed in Sodankylä before this measurement setup. We suggest to write this explicitly in the revised version of the manuscript.
* * *
**d) Technical corrections**

Author's reply:
We thank Referee#3 for his careful reading and time and effort in giving us these detailed technical corrections needed. We will implement these changes in the suggested revised version of the manuscript, as follows:

**l. 23: "OC/BC": please, define acronyms when used for the first time**: will do that

**l. 28: "Global Atmosphere Watch (GAW)":** ok

**l. 29: "World Meteorological Organization (WMO)":** ok

**l. 31: "International Polar Year (IPY) 2007-2008":** ok

**l. 39-43: at least one citation is needed here**: ok

**l. 53-54: define the "RT" acronym. This sentence is also quite confusing to the reader, since the quantity employed in radiative transfer models is effective albedo, not local albedo. Please, explain how the two quantities are related to each other.:** ok will do that. Referring to Meinander et al. (2008) we have earikler presented that "although the local albedo is affected by the regional albedo, our measurements at a height of 2 m may be considered to represent local albedo. Furthermore, a term "effective local albedo", for instance, could be more descriptive for the albedo quantity derived in our study. The critical question is whether the downwelling radiation field on the snow surrounding the observation point (i.e., in the area where the observed $F(\uparrow)$ originates), differs systematically from $F(\downarrow)$ at the obser-vation point. If not, $F(\uparrow/\downarrow)$ should be an accurate estimate of the local albedo."

**l. 126: add full stop at the end of the sentence:** ok

**l. 135: the SL501 erythemal irradiance is not "calculated" as a spectral integral, since broadband instruments cannot measure a spectrum and convolve it to an action spectrum. Rather, the measurement "represents" the convolution of the solar irradiance spectrum to the spectral response function of the instrument:** we thank the Referee#3 for this comment and will use the word represents.

**l. 144: why "spectral"?** the word spectral" was used, as the response of the sensor is with spectral weighting, although the outcome is one value.

**l. 151: is it essential to always specify "linux-computer"?** thanks, will avoid repeating the word "linux-computer" ultiple times.

**l. 154-160: please, describe only the differences between both systems, do not repeat the common characteristics:** agree and will avoid repeating and remove unnecessary repetition

**l. 176: are the sensors only "temperature controlled" or also "temperature stabilized"?** temperature stabilized is the correct term that will be used

**l. 182: "some data": please, explain what kind of data needed to be excluded:** ok, we suggest to add the following new explanation instead: "to avoid misinterpretation of data, knowledge of the SZA is essential because albedo changes according to the SZA. Again, the cosine response of the sensors affects the measurement results at low solar elevation angles. For data with SZA > 70 degrees, the cosine error is expected to increase dramatically. As most of the irradiance is then diffuse (at 300 nm more than 90%) this declines the impact of low Sun on the measurement results."

**l. 187: could you explain how you cope with the problem of tree shadows?** trees are cut if their shadows reach the albedo field.

**l. 212: "independent data of incoming and outgoing UV radiation": what do the authors mean by "independent"?** We refer to the fact that these are simultaneous measurement that can be used independently, as measured with two sensors separately, one upwards and one downwards. In opposite, some measurement systems are built so that the same sensor is used first down and then turned up, or two fixed optical heads are used one after another but their detector is the same, i.e., the measurement is done first up then down with one detector.

**l. 266: how is "c" defined?** We refer to Meinander et al. (2008) where we presented: *" a) Using the new re-calculated 1-min data for 4 January 2004, we calculated the simple SZA dependent empirical albedo decline, using a simple linear regression approach (albedo = f \* SZA). This slope f was calculated to equal with -0.0024; b) Using the original 8 minute-average-data, the decline during the day new slopes (f) were -0.002 for the afternoon data only, and -0.0028 for the whole day. In general, the Antarctic albedo was ranging from ~ 0.96–0.98 (0.98 in 1-min data, and 0.96 using 8-min data) to 0.86, resulting in a decline of ~0.10–0.12 (~10 %) towards the afternoon."*

-> we suggest to add here the description of "*defined using a simple linear regression approach (Meinander et al. 2008)*"

**l. 308:** "The Committee on Earth Observation Satellites (CEOS)": ok

**l. 330-331:** does this consideration apply to both "polar regions" (l. 328): thank you, the Referee#3 is right,it was Antarctic ozone loss in question here. We will add the word "Antarctic".
* * *
Helsinki, 22 June 2016

We have now given our replies to the comments of all the three Referees, as our replies were given first to Referee#1, then Referee#2, whereafter to Referee#3. If we have managed to reply satisfactorily all the concerns and suggestions of all the three Referees, we would like to suggest as the next step to prepare our major revised version of the manuscript for the consideration of all the three Referees, including all the changes suggested in our three replies. The language check will be made only to the final revised version of the manuscript.

Sincerely,
Outi Meinander, on the behalf of the co-authors